# Evaluating the risk of data loss due to particle radiation damage in a DNA data storage system

Christopher N. Takahashi[1], David P. Ward[1], Carlo Cazzaniga[2], Christopher Frost[2], Paolo Rech[3], Kumkum Ganguly[4], Sean Blanchard[4], Steve Wender[4], Bichlien H. Nguyen [1,5] ✉ & Jake A. Smith [1,5] ✉

DNA data storage is a potential alternative to magnetic tape for archival storage purposes, promising substantial gains in information density. Critical to the success of DNA as a storage media is an understanding of the role of environmental factors on the longevity of the stored information. In this paper, we evaluate the effect of exposure to ionizing particle radiation, a cause of data loss in traditional magnetic media, on the longevity of data in DNA data storage pools. We develop a mass action kinetics model to estimate the rate of damage accumulation in DNA strands due to neutron interactions with both nucleotides and residual water molecules, then utilize the model to evaluate the effect several design parameters of a typical DNA data storage scheme have on expected data longevity. Finally, we experimentally validate our model by exposing dried DNA samples to different levels of neutron irradiation and analyzing the resulting error profile. Our results show that particle radiation is not a significant contributor to data loss in DNA data storage pools under typical storage conditions.

The digital universe encompasses all digital information that is created globally and is expanding exponentially. By 2025, over 170 Zettabytes (ZB) of data will have been generated according to the International Data Corporation (IDC)[1]. While much of this data may be safely discarded, the scale of storage necessary to retain even the portion with enduring value overwhelms state-of-the-art archival systems utilizing magnetic tape storage media[2]. To meet future data storage demands, a scalable alternative storage media must be identified.

One such promising class of alternative storage media is oligonucleotides, typified by deoxyribonucleic acid (DNA) polymers, which have an information density on the scale of 1 Exabyte/mm$^3$ [3,4], many orders of magnitude greater than magnetic tape's estimated 37 Gigabits/mm$^3$ [5]. Early demonstrations have proven DNA a viable data storage medium, demonstrating the successful encoding of digital information into DNA sequences, writing of DNA sequences into

physical DNA polymers, reading of written DNA with sequencing, and decoding of the recovered sequences to recover the stored digital information[3,4,6–8]. With the scaling of these processes, DNA has the potential to replace magnetic tape as the media of choice in archival storage applications.

Archival storage media requires note just high information density but also longevity of the encoded data. On this metric, DNA also promises significant gains over magnetic tape and other conventional storage media, which have life spans in the tens of years[9]. To meet the demands of archival storage, where data is expected to be retained indefinitely, data thus must be re-copied onto new media at regular intervals, incurring economic and environmental costs[10,11]. DNA is inherently advantaged in this regard, having been shown to survive for millennia under the correct conditions. For example, encapsulation of DNA in dry, inert environments inspired by those found in naturally

[1]Paul G. Allen School of Computer Science and Engineering, University of Washington, Seattle, WA, USA. [2]Science and Technology Facilities Council, Swindon, UK. [3]University of Trento, Trento, Italy. [4]Los Alamos National Laboratory, Los Alamos, NM, USA. [5]Microsoft Research, Redmond, WA, USA. ✉e-mail: bnguy@microsoft.com; jakesmith@microsoft.com

occurring bone has been shown to yield DNA half-lives out to an estimated 50,000 years[12].

With data longevity identified as a primary design consideration in DNA data storage applications[13], the susceptibility of DNA to damage by water, light, and other environmental factors has each been previously studied[14]. These environmental factors are, however, routinely controlled in a data center environment[15]. Less easily controlled is the degree of exposure to ionizing particle radiation, a significant contributor to the relatively short life spans of magnetic storage media, where radiation damage has been implicated in material breakdowns and causes direct bit-flip errors in transistor-based media[16]. As exposure to ionizing radiation is known to damage biogenic DNA[17–19], it is necessary to consider if particle radiation exposure may have similarly deleterious effects on a DNA data storage system.

Here, we give consideration to the effects of particle radiation exposure on the life span of DNA media under prototypical storage conditions. We first calculate the rate of damage accumulation in a DNA data storage pool using mass action kinetics, approximating by Monte Carlo simulation cross sections for interactions of neutrons with both constituent nucleotides and the residual water molecules trapped with the DNA after dehydration. The sensitivity of the expected time to data loss due to particle radiation exposure is then explored across four parameters common to DNA data storage schemes: the number of strands a file is encoded across, the number of copies of each strand in the pool, the length of each strand, and the number of water molecules remaining after dehydration. Finally, the calculated values are experimentally validated by exposure of dried DNA samples to controlled levels of neutron irradiation in analogy to the particle radiation susceptibility tests commonly performed on magnetic media. Altogether, we find that particle radiation is not a substantial contributor to data loss in a typical DNA data storage pool.

## Results

### Composition of a DNA data storage pool

In a typical DNA data storage schema, groups of data-encoding sequences are written into DNA, amplified, and stored together in pools (Fig. 1). A given pool contains sequences of length $L$ encoding one or more digital files, with sequences encoding portions of the same file ($\{N_1, N_2, \ldots, N_f\}$) sharing a set of conserved primer regions to allow retrieval of a single file via selective amplification. The inter-primer region, the sequence of which encodes both a portion of the file and metadata to assist in file reassembly, is colloquially referred to as the payload.

In the theoretical extreme, a single copy of each comprising sequence $N_i$ is sufficient for storage of the file. In practice, however, DNA synthesis methods produce multiple copies of each sequence, giving the pool redundancy against damage to individual oligonucleotides. The mean number of oligonucleotides with sequence $N_i$ for a given file is known as the copy number ($C$) and may be readily increased prior to storage by polymerase chain reaction or similar. Data loss may be expected when the number of intact copies falls below the limit of detection of techniques for DNA retrieval and sequencing, which has been experimentally determined to be on the order of tens of copies[20]. We, therefore, set out to estimate the expected rate of strand loss in a DNA data storage pool due to particle radiation damage.

### Kinetics of particle radiation-induced DNA degradation

Due to the relatively large copy number $C$ in a typical DNA data storage pool, an unbiased estimate of the degradation kinetics may be determined assuming mass action kinetics. We begin by considering each reported mode of particle radiation-induced DNA damage. The most straightforward of these is the introduction of double-strand breaks by direct interaction of a particle with the DNA strand[21]. In the absence of cellular repair mechanisms, we treat the accumulation of such double-strand breaks as a series of independent, irreversible reactions:

$$C_n \xrightarrow{k_{\text{direct}}} D_n, \quad n = 1, 2, \ldots, N \tag{1}$$

where $C_n$ is the subpopulation of undamaged strands with index $n$ and $D_n$ is the corresponding subpopulation of damaged strands. In addition to double-strand breaks, such direct interactions have been shown to introduce base damage in neighboring nucleotides[22]. For the purpose of data longevity, however, we consider a double-strand break alone to be sufficient to prevent information recovery from the damaged strand and do not attempt to approximate these additional errors.

The rate constant $k_{\text{direct}}$ for accumulation of double-strand breaks by direct particle-DNA interactions is dependent on the time-averaged radiative flux ($\bar{\Phi}(E)$), the cross-section for the interaction of a particle with a single nucleotide residue ($\sigma_{\text{nuc}}(E)$), and the number of nucleotide residues in a strand ($L$), integrated over neutron energy $E$.

$$k_{\text{direct}} = L \int \bar{\Phi}(E)\sigma_{nuc}(E)dE \tag{2}$$

A second mode of particle radiation-induced DNA damage, known as quasi-direct, comes via the interaction of particles with water molecules in the immediate solvation sphere. These interactions result in the transfer of electrons and holes to neighboring DNA bases, culminating in base damage and strand breaks[23]. While this mode involves several potential reaction pathways[24], hole transfer from the hydration sphere is highly efficient[25], and the radical degradation reactions are likely to be relatively rapid. We, therefore, assume the generation of $H_2O^+$ by the interaction of a particle with water to be the rate-determining step and approximate the rate of DNA damage accumulation via this mechanism to be solely dependent on the concentration of hydration sphere water molecules ($[H_2O]$).

Given the close association required for electron and hole transfer from a water molecule to a DNA strand, we take $[H_2O]$ to include only those water molecules associated with undamaged strands $C_n$ and assume radiative damage to remain localized to the immediately associated strand. For convenience of implementation, we define a hydration ratio ($\theta$) to be the mean number of water molecules per nucleotide in the pool and restate the rate equation:

$$r = k_{quasi}[H_2O] = k_{quasi}\theta L[C_n] \tag{3}$$

Finally, we define $k_{\text{quasi}}$, the rate of interaction of hydration sphere water molecules with particles, as dependent on the cross-section for the interaction of a particle with a single water molecule ($\sigma_{H_2O}(E)$).

$$k_{quasi} = \int \bar{\Phi}(E)\sigma_{H2O}(E)dE \tag{4}$$

The last reported mode of particle radiation-induced DNA damage is indirect damage caused by solvent-derived radical species produced with the passage of particles through a solution-phase system[17,21,26]. We assume such indirect damage to be inconsequential given the dry storage conditions proposed in a typical DNA data storage schema and set $k_{\text{indirect}} = 0$[13].

Considering the rates of DNA damage due to the direct, quasi-direct, and indirect modes defined above, the net rate of damage accumulation in undamaged subpopulation $C_n$ is given by

$$\frac{d[C_n]}{dt} = -k_{direct}[C_n] - k_{quasi}\theta L[C_n] \tag{5}$$

The mean number of undamaged strands with index $n$ in a pool with copy number $C$ after exposure to radiative flux $\bar{\Phi}(E)$ for time $t$ is

File

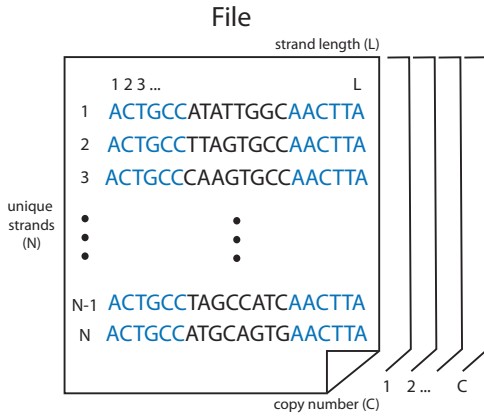

**Fig. 1 | Indexing in a DNA pool.** A given DNA pool comprises $N$ unique strands of length $L$, with $C$ copies of each strand $n$ providing physical redundancy. Each strand has a pair of 20 base primer regions (represented by the five shown in blue) flanking a payload (black).

then:

$$
\begin{aligned}
[C_n] &= [C]\exp\left(-t(k_{direct}+k_{quasi}\theta L)\right) \\
&= [C]\exp\left(-t\left(L\int\bar{\Phi}(E)\sigma_{nuc}(E)dE+\theta L\int\bar{\Phi}(E)\sigma_{H2O}(E)dE\right)\right)
\end{aligned} \quad (6)
$$

and the approximate half-life of strands in the DNA data storage pool:

$$
t_{1/2}=\log(2)\left(L\int\bar{\Phi}(E)\sigma_{nuc}(E)dE+\theta L\int\bar{\Phi}(E)\sigma_{H2O}(E)dE\right)^{-1} \quad (7)
$$

Radiative flux $\bar{\Phi}$ is dependent on the storage location and must in most cases be determined empirically. Parameters $L$ and $\theta$ are specific to a given DNA data storage schema and can in some cases be tuned toward optimal performance. In contrast, cross sections are fundamental to the respective molecules and translate across schema. Therefore, we next sought reliable estimates of $\sigma_{nuc}(E)$ and $\sigma_{H2O}(E)$ to enable efficient prediction of the half-life under a given schema.

## Estimation of interaction cross-sections

The relative quantities of observed particle radiation types are highly location-dependent[27]. We, therefore, elected to estimate $\sigma_{nuc}$ and $\sigma_{H2O}$ under terrestrial operating conditions at which a DNA data storage pool can be expected to encounter particle radiation primarily in the form of neutrons and muons. Here, the fluence of high-energy protons, electrons, and photons is minimized by atmospheric deflection, and the overall magnitude of particle flux is strongly correlated to elevation[28].

Studies on particle radiation-induced failures in electronic devices have demonstrated the rate of muon damage to be inconsequential relative to that of neutron damage[28]. The relative dearth of muon-induced damage is attributable to the muon's lack of strong interactions and, therefore, may be expected to translate to DNA data storage applications. With this in mind, we made an additional simplification, electing to estimate only the neutron-interaction cross sections $\sigma_{nuc}^n$ and $\sigma_{H2O}^n$ as approximations of the more general cross sections.

Neutron interaction cross sections are larger than those of protons and alpha particles, common secondary particles produced by the interaction of high-energy neutrons with the local environment, by approximately one and four orders of magnitude, respectively[29]. In the absence of heavy nuclei with which inelastic interactions result in a net increase in particles, this simplification, therefore, provides a conservative estimation of the total rate of particle radiation damage[30].

Where the effects of such materials are intended to be studied, the radiative flux function must be adjusted to compensate.

Finally, we selected a range of neutron energies between 10 and $1\times10^{11}$ eV on which to estimate the cross sections. This range generally covers non-elastic neutron-nuclei collisions, including the roughly defined resonance energy region, in which collisions produce energetically excited nuclei, through the fast and relativistic regions, in which collisions produce elementary particles[30].

**Sampling scheme.** A Monte Carlo sampling scheme was employed to estimate cross sections $\sigma_{nuc}^n$ and $\sigma_{H2O}^n$. To begin, the crystal structure of a dehydrated DNA duplex with sequence 5'-CGTGAATTCACG-3' and a hydration ratio of 3.83 residual water molecules per nucleotide was selected as representative due to the uniform distribution of bases in its composition[31]. The high level of symmetry innately present in double-stranded DNA ensures that the relative density of atoms is generally conserved across sequences, despite small differences in the exact atomic locations from sequence to sequence.

We hypothesized that the nucleotide-neutron interaction cross-section may be dependent on the thickness of the stored DNA due to the overlap of nuclei along the axis of irradiation. Therefore, a series of in silico samples were derived from the representative crystal structure with thicknesses ranging from 100 nm to 100 μm. First, explicit hydrogen atoms were added to the structure at interpolated locations using PyMol[32]. Next, the prepared crystal structure was stacked on the $xy$-plane, defining the $z$-axis to be the axis of irradiation, until the target thickness was reached. To approximate the assumed amorphous solid state, random rotations along the $x$-axis, $y$-axis, and $z$-axis drawn from $\mathcal{U}[0,2\pi]$ and random translations along the $x$-axis and $y$-axis drawn from $\mathcal{U}[-10\%,10\%]$ were applied to each unit cell prior to appending. Finally, the stack was projected onto the $xy$-plane and an area of 1 nm$^2$ was taken from the center for sampling.

At each energy, a series of $1\times10^8$ impact points on the $xy$-plane were drawn from the distributions $x\sim\mathcal{U}[-0.5,0.5]$ and $y\sim\mathcal{U}[-0.5,0.5]$ nm. For each impact point, the nearest nucleus, as determined by Euclidean distance, was identified, and the non-elastic neutron interaction cross-section of the nearest nucleus was estimated by interpolation of experimental cross-sections published by Brookhaven National Lab at the associated neutron energy[29]. The non-elastic cross section was approximated as the difference between the total and elastic cross sections. Linear interpolation was used within the bounds of experimental measurements, and a log-linear fit was used for extrapolation to higher particle energies. An interaction was recorded if the impact point lay within the cross-section of the nearest nucleus.

**Fitted curves.** Per-nucleotide and per-water interaction rates were tabulated across the sampled 1 nm$^2$ area to yield point estimates of the neutron interaction cross-sections at each simulated neutron energy. Interactions with neutrons having energies between $1\times10^6$ and $1\times10^{10}$ eV dominate the non-elastic cross sections. We, therefore, constructed a model with the general form of Eq. (8). Observations with neutron energies within this range were fit as a function of $\log_{10}(E)$ by Poisson regression, and observations with neutron energies outside this range were treated as marginal. Model coefficients for estimation of $\sigma_{nuc}^n$ and $\sigma_{H2O}^n$ are documented in Table 1. Cross sections are defined in units of barn and neutron energies in units of eV (Fig. 2).

$$
\log(\sigma^n)=\begin{cases}0 & \log_{10}(E)<6 \\ \beta_0+\beta_1\log_{10}(E)+\beta_2\log_{10}(E)^2+\beta_3\log_{10}(E)^3 & 6\leq\log_{10}(E)\leq10 \\ 0 & \log_{10}(E)>10\end{cases} \quad (8)
$$

Following our prior hypothesis, we additionally investigated the dependence of cross sections $\sigma_{nuc}^n$ and $\sigma_{H2O}^n$ on the thickness of the stored DNA along the irradiation axis. The simulated cross-section data

was refit to the general form of Eq. (8) with the addition of an exogenous term representing the thickness of the in silico samples prepared above. In both cases, no significant dependence of the observed cross section on DNA thickness was found ($\sigma_{nuc}^n$: $t(160) = -0.55$, $p = 0.59$) ($\sigma_{H2O}^n$: $t(160) = 0.32$, $p = 0.75$).

Finally, we considered the effect that GC-content may have on the observed cross-sections. It is feasible that the cross-section of a pair of deoxyguanidine and deoxycytidine residues may differ from a deoxyadenosine/deoxythymidine pair, given a slight difference in the identities and total number of constituent atoms. Similarly, it is feasible that these differences may extend to an altered interaction with water and, therefore, the observed cross section for water molecules in the hydration sphere. We, therefore, fit a third set of models, adding an exogenous term representing the fractional GC-content in the sampled area. Once again, no significant dependence of the observed cross section on GC-content was found in either case ($\sigma_{nuc}^n$: $t(160) = 1.08$, $p = 0.28$) ($\sigma_{H2O}^n$: $t(160) = -0.60$, $p = 0.55$).

## Information loss from particle radiation-induced damage

With estimates of $\sigma_{nuc}(E)$ and $\sigma_{H2O}(E)$ in hand, the accumulation of particle radiation-induced damage in a DNA data storage pool can be effectively modeled. We sought to complete the picture by tying the kinetics of DNA damage accumulation into a probabilistic model for system-wide loss of information.

In the simplest case, where data is encoded into the pool of DNA sequences without logical redundancy, information is lost when the number of existing strands $[C_n]$ for any $n \in \{1, N\}$ drops below the lower limit of detection ($C_{LLOD}$) for the sequencing method. A previous study empirically determined $C_{LLOD}$ to be approximately 10 copies using commonly employed Illumina sequencing and 15% logical redundancy[20]. This may be taken as a representative limit of detection for a typical DNA data storage scheme, but care must be taken to adjust the estimate to a specific scheme since the rate of damage accumulation slows as the number of undamaged strands approaches $C_{LLOD}$.

We begin by first approximating the time between damaging events as a series of exponentially distributed random variables $T_{[C_n]}$ such that the total time to information loss due to accumulation of damage to strands with index $n$ is given:

$$T_{loss}^n = \sum_{[C_n] = C_{LLOD}}^{C} T_{[C_n]}, \qquad T_{[C_n]} \sim \text{Exp}(\lambda([C_n])) \qquad (9)$$

where $\lambda([C_n])$ is the rate at which particle radiation-induced damage may be expected to accumulate in each subpopulation of strands $C_n$ as defined in the mass action kinetics model (Eq. (5)).

$$\lambda([C_n]) = \frac{d[C_n]}{dt} = [C_n]\left( L \int \bar{\Phi}(E)\sigma_{nuc}(E)dE + \theta L \int \bar{\Phi}(E)\sigma_{H2O}(E)dE \right) \qquad (10)$$

The time to information loss across the entire DNA data storage pool is then the minimum time to information loss across the constituent strands $C_n$:

$$T_{loss} = \min_{n \in \{1...N\}} T_{loss}^n \qquad (11)$$

In line with earlier assumptions, $T_{loss}^n$ is taken to be independent and identically distributed. The probability of information loss from the DNA data storage pool after storage time $t$ is then:

$$F_{T_{loss}}(t) = P(T_{loss} \le t) \qquad (12)$$

## Table 1 | Regression coefficients for estimation of cross-sections

|  | $\sigma_{nuc}^n$ |  | $\sigma_{H2O}^n$ |  |
|---|---|---|---|---|
|  | Coefficient | Std. deviation | Coefficient | Std. deviation |
| $\beta_0$ | −201.81377111 | 27.0 | −400.1662991 | 159 |
| $\beta_1$ | 72.93892997 | 10.4 | 144.52231949 | 59.9 |
| $\beta_2$ | −8.51321887 | 1.33 | −17.197389 | 7.46 |
| $\beta_3$ | 0.32227953 | $5.64 \times 10^{-2}$ | 0.67101674 | 0.308 |

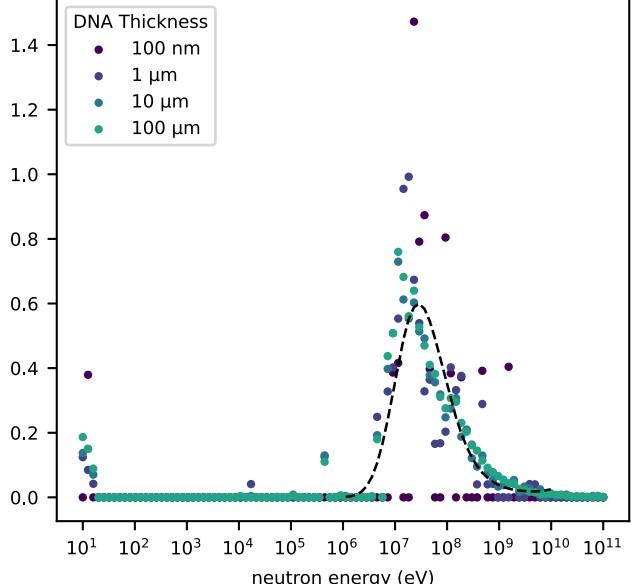

**Fig. 2 | Curve fitted simulation data.** Values for cross-sections $\sigma_{nuc}^n$ and $\sigma_{H2O}^n$ observed in the simulation are plotted as a function of neutron energy. The thickness of the simulated DNA is represented as a color gradient from purple (thinnest) to teal (thickness). Cross-sections are reported in barns, equivalent to $1 \times 10^{-28}$ m$^2$. The fitted models of cross-section as a function of neutron energy are overlaid in black dashes. Source data are provided as a Source Data file.

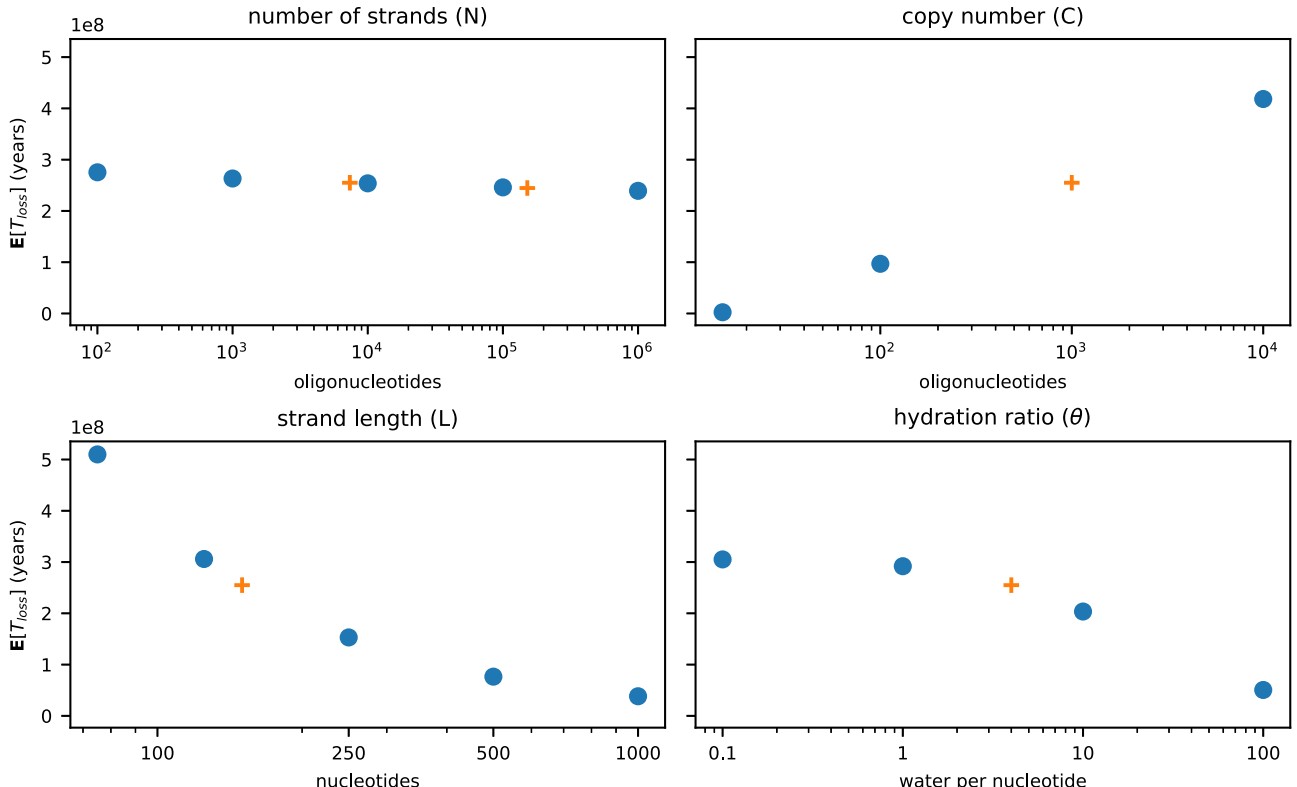

**Fig. 3 | Sensitivity of $E[T_{loss}]$ to changes in data storage scheme.** Expected values are calculated from a baseline of $N = 7373$, $L = 150$, $C = 1000$, and $\theta = 4$, with only the indicated variable changing at each point, and the JEDEC reference neutron spectrum[16]. Experimentally investigated parameters are presented as orange crosses. Source data are provided as a Source Data file.

$$= 1 - P(T_{loss}^n > t)^N \qquad (13)$$

$$= 1 - \left[1 - F_{T_{loss}^n}(t)\right]^N \qquad (14)$$

where $F_{T_{loss}^n}(t)$ is the cumulative density function of the hypoexponential distribution with parameters $\lambda \in \{\lambda(C)...\lambda(C_{LLOD})\}$, and the expected time to data loss due to particle radiation-induced damage may be evaluated:

$$\mathbf{E}[T_{loss}] = \int_0^\infty (1 - F_{T_{loss}}(t))dt \qquad (15)$$

Equation (15) may be used to understand the effect of several design parameters on the resiliency of a DNA data storage scheme to damage by particle radiation (Fig. 3). The number of strands across which a file is distributed ($N$), for example, has little effect on this resiliency, while the copy number ($C$) is observed to vary log-linearly with $\mathbf{E}[T_{loss}]$. Susceptibility to particle radiation damage increases with the length of each strand in the pool ($L$); however, the effect slows with increased $L$. Finally, $\mathbf{E}[T_{loss}]$ is relatively insensitive to the hydration ratio ($\theta$) in highly dehydrated samples but begins to degrade more quickly as the number of residual water molecules increases.

**Accelerated neutron irradiation experiments**
The validity of the model was tested by comparison to experimentally observed rates of damage accumulation in a set of DNA data storage pools exposed to controlled levels of neutron radiation. Two DNA-encoded files—a 115 kB JPEG image of a Space Shuttle and a 2.3 MB PDF topological map of the globe, encoded as previously described with 15% logical redundancy[3]—were selected for irradiation, amplified by

PCR, and serially diluted to copy numbers of $1 \times 10^3$ and $1 \times 10^6$. Each sample was concentrated to an amorphous film in a 0.5 mL Eppendorf tube by evaporation at room temperature in a laboratory fume hood.

Samples were irradiated with fast neutrons for up to three days on the Los Alamos Neutron Science Center (LANSCE) ICE 1 beamline (Fig. 4). A replicate set of samples was irradiated on the ChipIr beamline at the Rutherford Appleton Laboratory, UK. Negative controls failed for these replicates; therefore, all reported results are from the LANSCE irradiation experiments. Tested neutron fluences ranged from $1.6 \times 10^{11}$ to $5.1 \times 10^{11}$ neutrons per $cm^2$, with irradiation of the longest exposed samples equivalent to 4.4 million years of exposure at sea level in New York City. After irradiation, the samples were rehydrated in molecular-grade nuclease-free water, the concentration of intact strands determined by quantitative PCR, and the sequence of each determined by the Illumina method.

The qPCR data was first checked for evidence of radiation damage. Amplification curves were fit to a four-parameter logistic curve, and the cycle number corresponding to half the maximum amplification achieved ($C_{1/2}$) was determined. In each tested pool, no statistically significant correlation was observed between $C_{1/2}$ and the level of neutron radiation exposure (Fig. 5). As the per-cycle sensitivity of qPCR is $O(\log(2))$, the lack of a significant observed change in intact DNA concentration over the $4.4 \times 10^6$ year-equivalents of neutron irradiation is consistent with a half-life of $4.9 \times 10^7$ years ($\theta = 4$) as predicted by Eq. (7).

Next, the recovered sequences were checked for the introduction of single-nucleotide variants. Recovered sequences were aligned to the ground truth sequences, and the rates of deletion, insertion, and substitution errors were calculated on a per-base basis (Fig. 6). No statistically significant correlation was observed between the total error rate and the level of exposure ($t(29) = 1.1$, $p = 0.28$), nor was any statistically significant correlation observed for the rates of deletions,

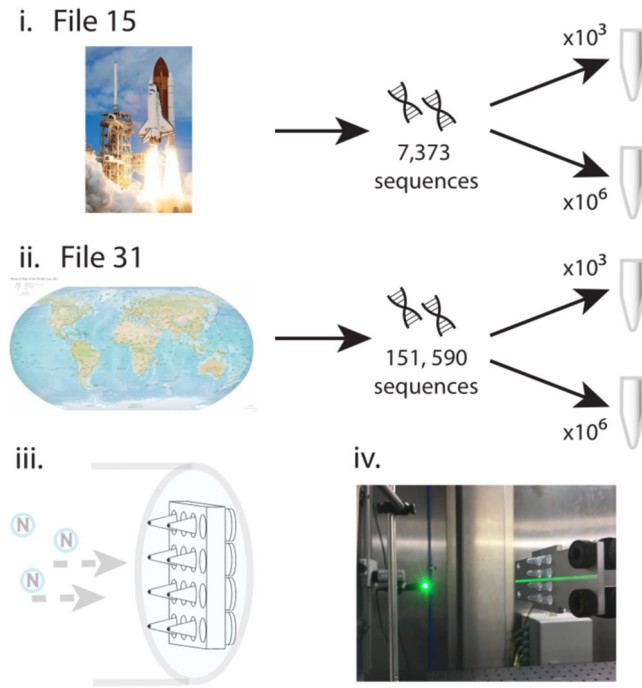

**Fig. 4 | DNA neutron experiment overview.** Two different sized files stored in DNA ((i) space shuttle and (ii) CIA world map) were placed in aliquots containing $10^3$ and $10^6$ copies of each file, respectively. The aliquots were then exposed to different fluence at LANSCE ICE. A schematic of the holder for the DNA samples and orientation in the beam (iii) and a typical mounting of DNA samples on the ChipIR beamline (iv) are shown. The Space Shuttle image is sourced from the NASA Image and Video Library. The CIA world map image is sourced from *The World Factbook* 2012.

insertions, or substitutions individually. The lack of a significant change in error rates with increasing exposure suggests that neutron radiation does not induce single-nucleotide variant damage at rates relevant to DNA data storage for the tested exposure, consistent with the assumptions made by the mass action kinetics model.

Finally, we attempted recovery of the encoded files. The recovered sequences were clustered and decoded with no knowledge of the ground truth sequences, and the resulting files compared to the originals. No bit errors were observed across all neutron radiation-exposure levels, consistent with the expected time to data loss calculated for each file as predicted by Eq. (15) (Fig. 3).

## Discussion

DNA is a naturally occurring polymer that has the potential to replace magnetic media in archival storage applications. In addition to allowing dramatically increased information density, DNA is more robust to data degradation than traditional media, making an attractive option from the perspective of both reliability and, due to elimination of data re-copy cycles, environmental sustainability.

Previous work on DNA preservation has established dramatic half-lives for properly stored DNA. In this work, we establish the role particle radiation-induced damage, a major contributor to data loss in magnetic media, plays in the loss of information from a DNA data storage pool. Our results demonstrate that dehydrated DNA, unlike magnetic media, is inherently resistant to damage from particle radiation. While we caution that the radiative flux function must be adjusted case-by-case to include the effect of secondary particles where quantitative predictions are desired, the presented model for expected data loss can be used to qualitatively understand trade-offs between susceptibility to particle radiation and several common design parameters in the creation of new DNA storage schemes.

## Methods

### Materials

Oligonucleotide pools for the encoded files were purchased through Twist Bioscience's oligo pool service. Forward and reverse primers were purchased from Integrated DNA Technologies. Sequences are provided via Zenodo at doi:10.5281/zenodo.12713629. Kapa HiFi enzyme mix (KK2601) was purchased from Roche. Eva Green dye (31000) was purchased from Biotinium. Illumina prep (20015965) and sequencing kits (20024908) were purchased from Illumina. AMPure XP beads (A63881) were purchased from Beckman Coulter. QIAquick PCR purification kits (28104) were purchased from QIAGEN. Molecular biology grade water (J71786.AP) was purchased from Fisher Scientific.

### DNA library preparation

All files were PCR amplified with the following protocol:
- 1 μL of ssDNA (~10 ng DNA)
- 0.5 μL of the appropriate forward primer at 10 μM
- 0.5 μL of the appropriate reverse primer at 10 μM
- 10 μL of 2 × Kapa HiFi enzyme mix
- 8 μL of molecular biology-grade water

The following PCR protocol (A) was used:
1. 95 °C for 3 min
2. 98 °C for 20 s
3. 62 °C for 20 s
4. 72 °C for 15 s
5. Repeat steps 2–4 until amplified (~15–25 cycles)

This protocol was scaled up via parallelization as needed to generate a high yield, at least 96 μg of DNA is recommended (1 μg per 96 ligation reactions). Subsequent sequencing preparation via ligation was done following Illumina Truseq Nano and ChIP ligation protocols. Briefly, samples were converted to blunt ends with the ERP2 reagent and then purified with AMPure XP beads. An 'A' nucleotide was added to the 3' ends of the blunt DNA fragments with A-tailing ligase, followed by ligation to the Illumina sequencing adapters with the TruSeq DNA CD Indexes. We then cleaned the samples with Illumina sample purification beads and enriched the sample using a 10-cycle PCR. Enrichment was done with the following protocol:
- 25 μL of DNA to be ligated
- 5 μL of the PCR Pimer Cocktail provided in the TruSeq Nano kit
- 20 μL of Enhanced PCR Mix provided in the TruSeq Nano kit

The following PCR protocol (B) was used:
1. 95 °C for 3 min
2. 98 °C for 20 s
3. 60 °C for 15 s
4. 72 °C for 30 s
5. Repeat steps 2–4 for 10 total cycles

In conclusion, each file was ligated to 96 unique Illumina indexes, and the final product length was verified using a QIAxcel Bioanalyzer. The final product (each of the 96 × 2 reactions) was then PCR purified using the QIAquick PCR Purification protocol and reagents to remove extraneous material.

All samples were then quantified using qPCR. Each file was amplified using the following qPCR recipe:
- 1 μL of 100× diluted DNA
- 1 μL of the Illumina forward/reverse adapter mix at 10 μM
- 10 μL of 2 × Kapa HiFi enzyme mix
- 7 μL of molecular biology grade water
- 1 μL of 20 × Eva Green

PCR protocol (A) was used to amplify the samples. Each sample was run in duplicate, and alongside an ultramer serial standard

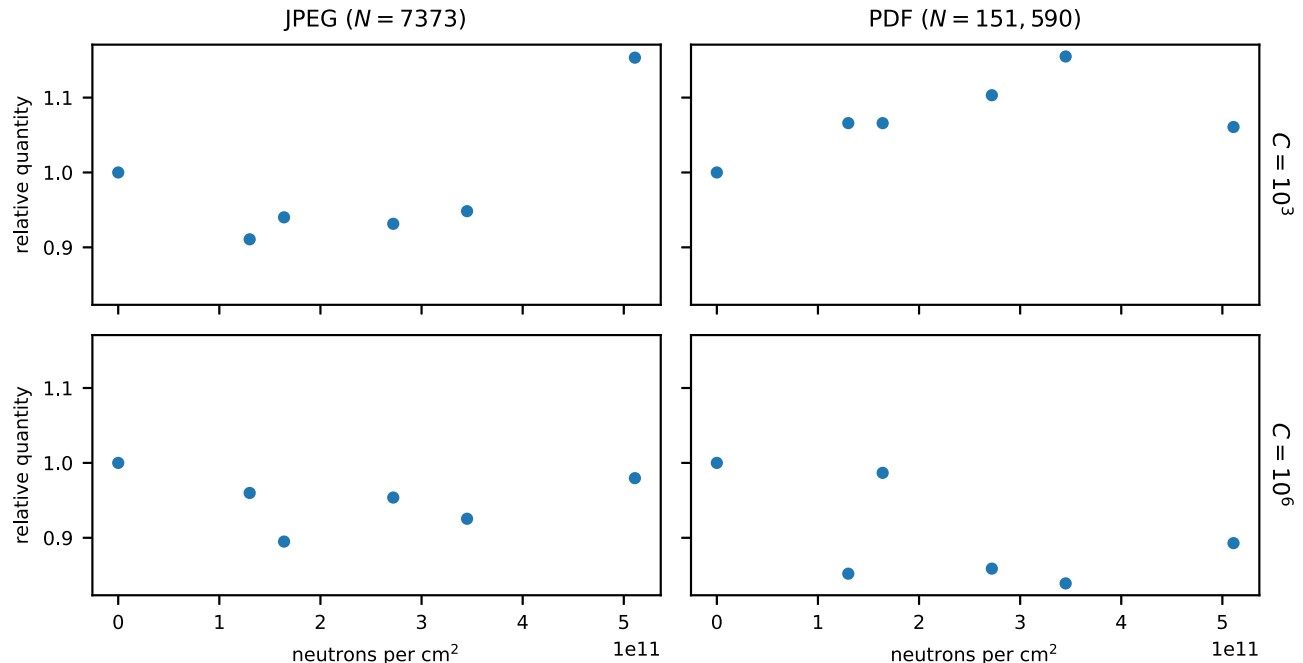

**Fig. 5 | The relative quantity of unbroken DNA remaining in each file as a function of neutron fluence.** DNA quantities were determined by qPCR with the resulting curves fit to a four-parameter logistic curve. No statistically significant correlation was observed (two-sided $t$-test, uncorrected acceptance criteria of $p < 0.05$) ($N = 7373$, $C = 10^3$: $t(4) = 1.49$, $p = 0.21$) ($N = 7373$, $C = 10^6$: $t(4) = -0.20$, $p = 0.85$) ($N = 151, 590$, $C = 10^3$: $t(4) = 1.3$, $p = 0.26$) ($N = 151, 590$, $C = 10^6$: $t(4) = -1.3$, $p = 0.26$). Source data are provided as a Source Data file.

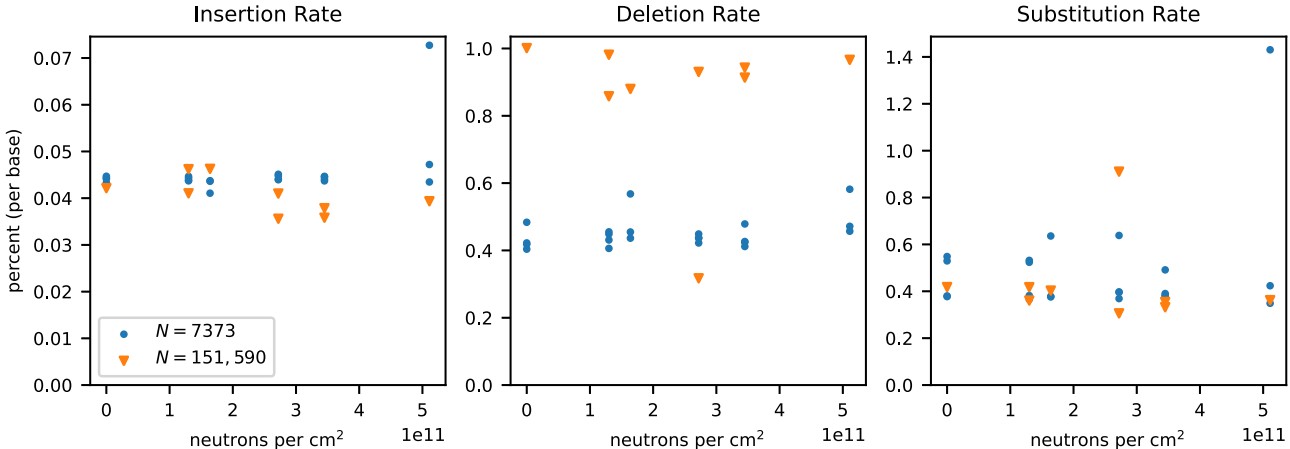

**Fig. 6 | The percentage of insertion, deletion, and substitution errors observed in each file as a function of neutron fluence.** Blue circles represent data points derived from experiments with $N = 7373$, and orange triangles from experiments with $N = 151,590$. Source data are provided as a Source Data file.

(ordered from IDT) to generate a standard curve. Samples were fit to this concentration curve via Quantstudio Design and Analysis Software.

The mass $m$ of DNA present in each tube is

$$m = 100(cV_0) \qquad (16)$$

where $c$ is concentration, $V_0$ is template volume, and 100 is the dilution factor. In order to increase the odds of generating observable levels of single-nucleotide variants, we aimed to perform the experiment at the lowest possible level of physical redundancy. Number of copies $N_c$ in each sample is

$$N_c = \frac{6.022 \times 10^{23}\, m}{(310)(650)(1 \times 10^9)} \qquad (17)$$

Each sample was diluted and aliquotted to yield the desired experimental copy number, either $1 \times 10^3$ or $1 \times 10^6$ copies of each of the sequences. Each sample was transferred to a 0.2 mL PCR tube and dried overnight in a fume hood, leaving an amorphous DNA film concentrated at the bottom of the tube. Samples were capped and shipped to LANSCE for neutron irradiation experiments. Post-exposure, samples were shipped back once they had cooled down. Shipping controls for each file were included in the study and represent zero fluence exposure.

**DNA sequencing**
Upon return, the dry neutron radiation-exposed DNA was rehydrated in 20 μL of molecular-grade nuclease-free water and quantified using qPCR (method described above). Low-copy number samples were enriched for sequencing post-radiation exposure. The neutron radiation-exposed samples were mixed into sequencing libraries

proportionally (e.g., a 7000 oligonucleotide sequence to be sequenced with a 150,000 oligonucleotide sequence would make up 4.4% of the total material) and prepared for sequencing by following the Illumina NextSeq Denature and Dilute Libraries Guide. Sequencing libraries were loaded in the Illumina NextSeq at 1.3 pM with a 20% control spike-in of ligated PhiX genome.

### Statistical tests

All reported statistical tests were two-sided $t$-tests that fit with the statsmodels Python package[33]. An uncorrected acceptance criterion of $p < 0.05$ was used for each test, and implicit assumptions of normality were made. With the exception of the qPCR curves shown in Fig. 5, measurements were taken from distinct samples.

### Reporting summary

Further information on research design is available in the Nature Portfolio Reporting Summary linked to this article.

### Data availability

The DNA sequences comprising the encoded files and raw sequencing reads generated in this study have been deposited in the Zenodo database under the accession code https://doi.org/10.5281/zenodo.12713629. The irradiation count, error rate, qPCR, and Monte Carlo simulation data generated in this study are provided in the Supplementary Information/Source Data file. Source data are provided in this paper.

### Code availability

Code necessary to reproduce the estimation of interaction cross-sections is provided in the supplementary information.

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

### Acknowledgements

The authors acknowledge Yuan-Jyue Chen and Karin Strauss for useful discussion. The authors additionally acknowledge Luis Ceze and the Molecular Information Systems Laboratory for contributing lab space. This work was funded by Microsoft Corporation.

## Author contributions

J.A.S. performed kinetic modeling. J.A.S. and C.N.T. performed statistical modeling. D.P.W. and B.H.N. performed DNA sample preparation, amplification, and sequencing. B.H.N., C.C., C.F., P.R., K.G., S.B., and S.W. designed and performed neutron radiation exposure experiments. J.A.S., C.N.T., and S.B. performed data analysis. J.A.S., C.N.T., and B.H.N. wrote and edited the manuscript. B.H.N. and C.N.T. conceived the project.

## Competing interests

J.A.S. and B.H.N. were employed by Microsoft for the duration of this work. The remaining authors declare no competing interests.
