## [Peer Review File · Nature Communications]

Reviewers' Comments:

Reviewer #1:

Remarks to the Author:

This is a very interesting article describing the impact of neutron-induced DNA damage on a DNA data-storage system involving theoretical and experimental analysis of the neutron-induced DNA damage.

I am from the health physics domain and I have an understanding of radiation-induced DNA damage in the in vivo context. I am not familiar with the DNA data-storage domain and thus I cannot comment on the novelty of this research in that domain. What I can do is provide my perspective from the health physics domain on the biophysics encountered when considering radiation-induced DNA damage.

The described DNA data-storage system has a number of important differences to in vivo DNA. Most importantly, the DNA in the data-storage system is dehydrated, which allows the indirect DNA damage route to be neglected as the authors have done. Were water present, as in in vivo systems, the DNA damage would surely have been more significant.

One potentially important consideration that I find missing from the article is the effect of secondary particles. The authors consider primary neutrons as their irradiation source and primary neutron-induced DNA damage as their form of damage. This is presumably motivated by the neutron-induced failures that have been observed in electronic devices (as per line 253). However, neutron-induced DNA damage should not be considered as similar to neutron-induced electronic damage. In the in vivo context, neutron-induced DNA damage is not typically due to either direct or indirect action of primary neutrons themselves. Rather, it is due to the action of the secondary particles that are produced when the primary neutrons interact upstream in the body.

Without knowing the physical details of the data-storage system, it is difficult to comment on how secondary particles would impact the DNA. As such, I feel that the authors should provide a description of their data-storage facility and not just the individual data-storage pool that was bathed with neutrons. For long-term storage, will the DNA film be stored with multiple films sitting on top of each other or will there be shielding between each film or will individual films be stored in air or deep underground? Knowing how the films are packaged and how they will be stored will allow for Monte Carlo modelling of upstream neutron interactions such that the DNA damage caused by secondary particles moving downstream can be considered. Unless the storage conditions are modelled (or experimentally irradiated) with full secondary particle considerations, the neutron-induced DNA damage described by the authors should only be considered theoretical and best-case. I believe this limitation of the work should be mentioned in the paper. Likewise, realistic DNA data-storage facilities would likely experience a diversity of radiation from naturally occurring nuclides. These should also be considered for any realistic understanding of the long-term impact of radiation on DNA data storage.

Reviewer #2:

Remarks to the Author:

Review on manuscript titled

"Evaluating the Risk of Data Loss Due to Particle Radiation Damage in a DNA Data Storage System"
by Takahashi et al.

Submitted to Nature Communications

The manuscript aims to investigate the effect of particle radiation, mainly due to neutron interactions, on information stored in DNA. The authors first develop an estimated kinetics model to simulate the influence of particle interactions on DNA damage due to presence of H₂O.⁺ and then test the model experimentally via a controlled neutron exposure radiation, on two JPEG and PDF files stored in DNA using the common encoding and synthesis methods.

Overall, I find this manuscript scientifically sound, in terms of rationale, assumptions made to create the kinetics model and the way they have simplified the problem to study a problem that has been discussed within the community for long. The text is well-written, and paradigms are well-explained, and I must say I enjoyed reading the article and see it as a valuable piece of work. Still, I am not sure whether we can categorize this work as “specialized” or “of interest for the general scientific community”. I do believe that it could belong to the second category, which Nature Communications does majorly support and publish if the authors include more details and slightly revise the manuscript so the general audience connects with it a bit more.

Having said that, I'd like to ask a few questions and leave some comments:

Questions:

1- Under “Sampling Scheme”, the authors mention that a model DNA sequence composed of CGTGAATTCACG is selected to study. Any thoughts on whether the palindromic nature of this sequence would contribute to lower/higher levels of exposure to water molecule radicals? A related question would be, is there a way to determine that the uniform distribution of bases could actually contribute to the level of damage on DNA?

2- Consecutive bases are usually avoided in data encoded DNA, particularly to prevent structures such as G-quadruplexes. Any thoughts on whether such structures could contribute/lessen the damage?

3- Figure 6 shows the percentage of errors occurring in each file after exposure. The data does show that the effect is insignificant. It is however interesting that in the middle panel (deletion rate), the larger file has a ~2-times more deletion per base compared to the small file. I'm curious if the authors have any thoughts on the reason.

Comments:

- I think figure designs and plots could be enhanced so the general audience can appreciate the content more. Fonts are inconsistent and, in some cases, too small to read.
- Under Methods, DNA Library Preparation, the first PCR protocol, it'd be nice if authors include the number of cycles used (or a just a range) to achieve 1 μ g DNA per reaction.
- In some cases, the name of product manufacturers is not listed.
- Page titled “Appendix A” is empty. Not sure if anything's meant to be there.

Our thanks to both reviewers for taking the time to read and provide constructive feedback on our research. Please find responses to the specific comments you have made below.

Reviewer #1 (Remarks to the Author):

This is a very interesting article describing the impact of neutron-induced DNA damage on a DNA data-storage system involving theoretical and experimental analysis of the neutron-induced DNA damage.

I am from the health physics domain and I have an understanding of radiation-induced DNA damage in the in vivo context. I am not familiar with the DNA data-storage domain and thus I cannot comment on the novelty of this research in that domain. What I can do is provide my perspective from the health physics domain on the biophysics encountered when considering radiation-induced DNA damage.

The described DNA data-storage system has a number of important differences to in vivo DNA. Most importantly, the DNA in the data-storage system is dehydrated, which allows the indirect DNA damage route to be neglected as the authors have done. Were water present, as in in vivo systems, the DNA damage would surely have been more significant.

One potentially important consideration that I find missing from the article is the effect of secondary particles. The authors consider primary neutrons as their irradiation source and primary neutron-induced DNA damage as their form of damage. This is presumably motivated by the neutron-induced failures that have been observed in electronic devices (as per line 253). However, neutron-induced DNA damage should not be considered as similar to neutron-induced electronic damage. In the in vivo context, neutron-induced DNA damage is not typically due to either direct or indirect action of primary neutrons themselves. Rather, it is due to the action of the secondary particles that are produced when the primary neutrons interact upstream in the body.

This is an important point that we have failed to adequately address in our original draft; thank you for raising it. We considered inclusion of secondary particle interactions within the study, but ultimately elected to approximate the entire radiative flux of particles as neutrons. The interaction cross sections reported within the Brookhaven database for protons and alpha particles are roughly one and four orders of magnitude smaller, respectively, than those of neutrons. Given this disparity, the all-neutron approximation leads to a conservative estimate of DNA damage at a given flux.

We have added language to the section "Estimation of Interaction Cross Sections" directly presenting this assumption and provided a disclaimer that the rate of radiative flux will need to be adjusted where heavy nuclei are included in the storage scheme.

Without knowing the physical details of the data-storage system, it is difficult to comment on how secondary particles would impact the DNA. As such, I feel that the authors should provide a description of their data-storage facility and not just the individual data-storage pool that was bathed with neutrons. For long-term storage, will the DNA film be stored with multiple films

sitting on top of each other or will there be shielding between each film or will individual films be stored in air or deep underground? Knowing how the films are packaged and how they will be stored will allow for Monte Carlo modelling of upstream neutron interactions such that the DNA damage caused by secondary particles moving downstream can be considered. Unless the storage conditions are modelled (or experimentally irradiated) with full secondary particle considerations, the neutron-induced DNA damage described by the authors should only be considered theoretical and best-case. I believe this limitation of the work should be mentioned in the paper. Likewise, realistic DNA data-storage facilities would likely experience a diversity of radiation from naturally occurring nuclides. These should also be considered for any realistic understanding of the long-term impact of radiation on DNA data storage.

At the current state of the field, there is unfortunately not a de facto standard for the form factor of a DNA data storage system. It seems very clear from studies on hydrolytic susceptibility that the DNA will be stored in a dehydrated state, but little else has been firmly decided. With this publication, we aim to provide information on a damage mechanism commonly considered in the traditional data storage world, culminating in a set of guidelines as shown in Figure 3 that can be used in the eventual design of such a form factor.

Given this uncertainty, we agree that the present work should be considered less as a strict quantitative analysis and more on the strength of the qualitative guidance it provides. To this end, we take care to acknowledge the underlying modeling assumptions throughout the text and have added language to the conclusion to acknowledge this limitation.

Reviewer #2 (Remarks to the Author):

Review on manuscript titled

“Evaluating the Risk of Data Loss Due to Particle Radiation Damage in a DNA Data Storage System”
by Takahashi et al.

Submitted to Nature Communications

The manuscript aims to investigate the effect of particle radiation, mainly due to neutron interactions, on information stored in DNA. The authors first develop an estimated kinetics model to simulate the influence of particle interactions on DNA damage due to presence of H₂O.⁺ and then test the model experimentally via a controlled neutron exposure radiation, on two JPEG and PDF files stored in DNA using the common encoding and synthesis methods. Overall, I find this manuscript scientifically sound, in terms of rationale, assumptions made to create the kinetics model and the way they have simplified the problem to study a problem that has been discussed within the community for long. The text is well-written, and paradigms are well-explained, and I must say I enjoyed reading the article and see it as a valuable piece of work. Still, I am not sure whether we can categorize this work as “specialized” or “of interest for

the general scientific community". I do believe that it could belong to the second category, which Nature Communications does majorly support and publish if the authors include more details and slightly revise the manuscript so the general audience connects with it a bit more. Having said that, I'd like to ask a few questions and leave some comments:

Questions:

1- Under "Sampling Scheme", the authors mention that a model DNA sequence composed of CGTGAATTCACG is selected to study. Any thoughts on whether the palindromic nature of this sequence would contribute to lower/higher levels of exposure to water molecule radicals? A related question would be, is there a way to determine that the uniform distribution of bases could actually contribute to the level of damage on DNA?

In a more complex model of DNA stability including UV irradiation, temperature, and other damage vectors, we would clearly expect a sequence dependence. Here, where we aim to isolate the effects of particle radiation as a more difficult to control environmental variable, the primary potential cause of sequence dependence is likely the one you identify: correlation between the sequence and the location of residual water molecules.

In defining a "hydration ratio" of water molecules per nucleotide, our model effectively abstracts this potential correlation with a mean-field assumption. We believe this to be a fair abstraction to make as long as the modeled quantity remains number of damaged strands. A model of damage on the per-nucleotide basis would clearly need to take the effect of local sequence into account.

With regards to the more general question of base distribution, we considered several crystal structures available in the Protein Data Bank, with the CGTGAATTCACG sequence eventually being chosen as the model sequence primarily due to the inclusion of the close-packed water molecules in the structure, which we deemed critical to representation of the dehydrated storage state. You are correct that this choice is not without bias, though. In particular, clustering of GC residues around the tails of the sequence results in their underrepresentation in the model system (the mean GC content was 33.5%), as they are more likely to fall in the truncated outer region.

It was for this reason that we evaluated the significance of the GC fraction on the fitted neutron interaction cross sections. The lack of an observed significant effect is not particularly surprising, given the atomistic treatment of the nucleotides during the Monte Carlo simulations. As collections of nuclei in space, there is not a tremendous difference between the respective pyridine and pyrimidine bases.

2- Consecutive bases are usually avoided in data encoded DNA, particularly to prevent structures such as G-quadruplexes. Any thoughts on whether such structures could contribute/lessen the damage?

Pointing back to our thoughts above, there is not a tremendous difference between the individual pyridine and pyrimidine bases at the scale of particle radiation. We therefore believe that sequence-dependent particle radiation damage is unlikely to be observed.

3- Figure 6 shows the percentage of errors occurring in each file after exposure. The data does show that the effect is insignificant. It is however interesting that in the middle panel (deletion

rate), the larger file has a ~2-times more deletion per base compared to the small file. I'm curious if the authors have any thoughts on the reason.

The deletion rate is undoubtedly higher in the larger file. As the higher rate is observed in the control samples, we would attribute it to either the synthesis or amplification processes. Grossly speculating, this may be the result of a systematic inefficiency in the DMT-deprotection step as the rates of insertion and substitution errors are not significantly affected.

Comments:

- I think figure designs and plots could be enhanced so the general audience can appreciate the content more. Fonts are inconsistent and, in some cases, too small to read.

Our apologies, we have standardized font sizes and figure widths across the figures. The plot of the normalized LANSCE ICE neutron spectrum and measured neutron flux at Los Alamos, previously Figure 4b, was unable to be replotted in the same style, as it was taken from the LANSCE ICE testing manual. We have moved it to the supplementary information as a result.

- Under Methods, DNA Library Preparation, the first PCR protocol, it'd be nice if authors include the number of cycles used (or a just a range) to achieve 1ug DNA per reaction.

Our apologies, we have added additional information on the PCR protocol to the associated methods section.

- In some cases, the name of product manufacturers is not listed.

Our apologies, we have added a materials section to the Methods.

- Page titled "Appendix A" is empty. Not sure if anything's meant to be there.

Thanks for raising this. The Extended Data appendix is part of the latex template, but we have not utilized it for this submission. We have gone ahead and removed it to avoid confusion.

Reviewers' Comments:

Reviewer #1:

Remarks to the Author:

The corrections made in response to my review are adequate. The authors now appropriately acknowledge the issue of secondary particles and associated limitations.

Reviewer #2:

Remarks to the Author:

I appreciate the authors' response to my comments. I have no further questions and support the publication of the article in its revised format.